# Solid-State Fermentation of Sorghum by *Aspergillus oryzae* and *Aspergillus niger*: Effects on Tannin Content, Phenolic Profile, and Antioxidant Activity

**DOI:** 10.3390/foods11193121

**Published:** 2022-10-07

**Authors:** Pilar Espitia-Hernández, Xóchitl Ruelas-Chacón, Mónica L. Chávez-González, Juan A. Ascacio-Valdés, Antonio Flores-Naveda, Leonardo Sepúlveda-Torre

**Affiliations:** 1Bioprocess and Bioproducts Research Group, Food Research Department, School of Chemistry, Autonomous University of Coahuila, Saltillo 25280, Coahuila, Mexico; 2Departamento de Ciencia y Tecnología de Alimentos, Universidad Autónoma Agraria Antonio Narro, Buenavista, Saltillo 25315, Coahuila, Mexico

**Keywords:** hydrolyzable tannins, condensed tannins, ABTS, DPPH, HPLC-MS

## Abstract

Sorghum contains antioxidants such as tannins. However, these are considered antinutritional factors since they are responsible for the low digestibility of proteins and carbohydrates. Nevertheless, these can be extracted by solid-state fermentation (SSF). Therefore, this study aimed to evaluate the effects of SSF from *Aspergillus oryzae* and *Aspergillus niger* Aa210 on the tannin contents, phenolic profiles determined by HPLC-MS, and antioxidant activities (ABTS, DPPH, and FRAP) of two genotypes of sorghum. The results showed that with SSF by *A. niger* Aa210, a higher tannin content was obtained, with yields of 70–84% in hydrolyzable tannins (HT) and 33–49% in condensed tannins (CT), while with SSF by *A. oryzae* the content of HT decreased by 2–3% and that of CT decreased by 6–23%. The extracts fermented by *A. niger* at 72 and 84 h exhibited a higher antioxidant activity. In the extracts, 21 polyphenols were identified, such as procyanidins, (+)-catechin, (-)-epicatechin, scutellarein, arbutin, and eriodictyol, among others. Therefore, SSF by *A. niger* was an efficient process for the release of phenolic compounds that can be used as antioxidants in different food products. It is also possible to improve the bioavailability of nutrients in sorghum through SSF. However, more studies are required.

## 1. Introduction

Sorghum (*Sorghum bicolor* (L.) Moench) is the fifth most important cereal in the world and is considered a food crop of great nutritional value. It is a gluten-free cereal, which can benefit people with celiac disease [1]. Sorghum is the main cereal grains with great nutritional and functional potential, numerous studies on sorghum for human consumption have been undertaken [2]. The most outstanding group of bioactive compounds in sorghum is polyphenols, mainly phenolic acids, flavonoids, and tannins [3]. These compounds are the ones that mainly contribute to the antioxidant capacity of sorghum grains. The consumption of sorghum foods with significant amounts of polyphenols may have a beneficial effect in reducing the risk of chronic diseases, such as breast and colon cancer and diabetes [4]. Although sorghum has adequate contents of nutrients, its use in food has been limited, mainly due to the content of condensed tannins, considered antinutritional factors, since these interfere with the digestibility of proteins and carbohydrates, decreasing the absorption of nutrients [5]. However, these compounds in sorghum are those that have higher levels of antioxidants than in any other cereal [6].

Cereal processing can decrease or increase bioactive compound levels and therefore modify their bioavailability. The solid-state fermentation (SSF) process is a way to improve the phenolic contents and antioxidant properties in fermented foods [7]. Fermentation by microorganisms is also recognized for its efficiency and respect for the environment and has been reported to promote the effective release of phenolic acids from materials such as soybean products, rye bran, finger millet, and whole rye sourdoughs [8]. Some important factors that should be considered for the SSF process involve the correct selection of the substrate, moisture, and microorganisms. Fungi and yeasts have been called appropriate microorganisms for SSF [9]. The genus *Aspergillus* comprises various species of filamentous fungi, most of which are of great importance for industrial applications due to their high amylolytic capacity in the degradation of plant cell wall polysaccharides. In addition, their products have obtained a GRAS (generally regarded as safe) status [10]. Considering that SSF is an alternative for the release of bioactive compounds from this important cereal, the aim of this research was to evaluate the effect of SSF by the fungi *Aspergillus oryzae* and *Aspergillus niger* on the releases of tannin content, phenolic profiles, and antioxidant activities of the fermentative extracts of grains of two sorghum genotypes. These important findings may provide useful information on SSF by *A. niger* as an efficient bioprocess for the release of phenolic compounds from sorghum grains that can be used as antioxidants in different food products as well as on improving the bioavailability of nutrients in sorghum by decreasing the content of tannins through SSF by *A. oryzae*. However, more studies are required in this regard.

## 2. Materials and Methods

### 2.1. Genetic Material, Microorganisms, and Chemicals

Two pigmented sorghum genotypes were studied, red color (LES 5) and black color (GB), which are free-pollinated varieties of the Autonomous Agrarian University Antonio Narro. The *Aspergillus niger* Aa210 and *Aspergillus oryzae* strains used in this study belong to the collection of the Food Research Department of the Autonomous University of Coahuila. The following chemical reagents were used: bromocresol green (≥100%), methyl red (≥100%), boric acid (≥100%), SeO_2_ (99%), K_2_SO_4_ (≥99%), CuSO_4_.5H_2_O (≥98%), Na_2_CO_3_ (≥100%), HCl (37%), ethanol (96%), and K_2_S_2_O_8_ (≥99%) purchased from Chemical Industrial Technology (Nuevo Leon, Mexico); NaOH (99.99%) and H_2_SO_4_ (98%) purchased from SUMILAB (Coahuila Mexico); Folin–Ciocalteu reagent (2.0 N), gallic acid (97.5%), catechin (≥96%), DPPH (1,1-diphenyl-2-picrylhydrazyl) (97%), ABTS ^+^ (2, 2 ‘-azino-bis (3-ethylbenzothiazoline-6-sulfonic acid)) (≥98%), TPTZ (2,4,6-tris(2-pyridyl)-s-triazine) (≥98%), Trolox (97%), petroleum ether (≥90%), formic acid (≥95%), and acetonitrile (≥99.93%), which were purchased from Sigma-Aldrich (Missouri, USA); sodium acetate anhydrous (99%), glacial acetic acid (99.5%), and filter paper No. 1 purchased from Favela Pro, S.A. de C.V. (Sinaloa Mexico); iso-butyl alcohol (99%) reagent and methanol (99.8%) purchased from Jalmek Scientist (Nuevo Leon, Mexico); and (NH_4_) Fe (SO_4_)_2_ (98.5%) purchased from Karal (Guanajuato, Mexico).

### 2.2. Sample Preparation

The grain was cleaned by aeration and washed with purified water to remove all physical impurities. The sorghum grains were dried at room temperature for 24 h and dried in an oven for 48 h at 55 °C to remove excess moisture. Sorghum grains of each genotype were milled separately using a grain mill (The Blendtec Kitchen Mill, Orem, UT, USA).

### 2.3. Proximal Physicochemical Characterization

The moisture content was determined on a stove at a temperature of 130 °C for 2 h [11]. The water absorption index (WAI) and critical humidity point (CHP) tests were determined according to the methodology described by Buenrostro et al. [12]. For the WAI, 1.5 g sorghum samples were taken and placed in 50 mL conical Falcon tubes, and 15 mL of distilled water was added. The samples were shaken for 1 min at room temperature then centrifuged at 3000 rpm for 10 min. The supernatant was decanted, and the WAI was calculated considering the weight of the gel and was expressed as g of gel/g of dry weight. The CHP was determined with the help of a humidity analyzer (Precisa, model XM 50) by placing 1 g of a gel sample resulting from the previous test at a temperature of 120 °C until it reached a constant weight.

The protein content was estimated with nitrogen determination by the Kjeldahl method [13], and the results were multiplied by a factor of 6.25. For the analysis of lipids, 0.5 g samples were used in sealed XT4 filter bags for the automated Soxhlet method using an ANKOM XT15 fat analyzer following the AOCS method Am 5-04 procedure [14]. Petroleum ether was used as a solvent to perform the extraction. The lipid content was determined by weight loss. The crude fiber was analyzed from 1 g samples in filter bags (ANKOM F57) sealed using an ANKOM 2000 fiber analyzer following the AOCS method Ba 6a-05 procedure [15]. Previously, the bags were placed in a beaker with petroleum ether. The digestion of the samples was conducted with solutions of H_2_SO_4_ 0.255 N and NaOH 0.313 N. The fiber content was calculated by weight loss. The ash content was determined in 2 g of sample according to the AOAC method [13] in a Thermo Scientific Lindberg Blue M muffle at 600 °C for 2 h. The mineral content was determined by the elemental sensitivity method using an epsilon 1 X-ray fluorescence analyzer.

The contents of the available carbohydrates and sugars were calculated by the following formula:HC = 100 − (moisture + protein + lipids + ash + crude fiber)(1)

### 2.4. Solid-State Fermentation (SSF)

The fungal strains of *A. oryzae* and *A. niger* A210 were activated in PDA culture medium in Petri dishes at an incubation temperature of 30 °C for 5 days. A radial growth test of each fungus was carried out as a test of the adaptation of the microorganisms to the substrate, which was measured in triplicate, estimating the increase in the diameter of the microorganism colony at four points, using 3 g of sorghum as a carbon source and energy in Petri dishes with diameters of 9 cm and 8 mm explants of the microorganisms. Consistent moisture (70%) and incubation (30 °C) conditions were used, recording data every 12 h for a total time of 72 h.

The SSF was established in Petri dishes with 3 g of sorghum sample. All the treatments were performed for each fungus in triplicate. The conditions used were a 30 °C temperature, 70% moisture, and inoculum doses of 1 × 10^6^ spores/g of the sample (it was adjusted with the help of a Neubauer chamber). For pH determination, 1 g of sample was homogenized in 10 mL of distilled water at room temperature for 10 min and was subsequently measured using a potentiometer (METTLER TOLEDO Seven Easy, Greifensee, Switzerland).

The fermentation extracts were recovered with 15 mL of ethanol (96%) by manual pressing and filtered with No. 1 filter paper; they were obtained every 12 h for a total time of 96 h.

### 2.5. Determination of Phenolic Content

The hydrolyzable tannin (HT) contents of the extracts was determined using the Folin–Ciocalteu reagent according to the methods described by Wong et al. [16]. First, 20 μL of the extract was placed in a microplate mixing 20 μL of Folin–Ciocalteu reagent. After 5 min, 20 μL of sodium carbonate (0.01 M) was added, allowing it to react for 5 min. Finally, the solution was diluted with 125 µL of distilled water, and its absorbance was read at 790 nm in a microplate reader (Epoch, Biotek, Instruments, Inc., Santa Clara, CA, USA). Values were calculated using a gallic acid calibration curve (0–700 ppm). The results were expressed as mg gallic acid equivalents (GAE)/g sample. The quantification of condensed tannins (CT) was determined using the HCl-butanol technique by Sepúlveda et al. [17]. Briefly, 0.5 mL of extract was placed in test tubes. Then, 3 mL of HCl-terbutanol (1:9) and 0.1 mL of ferric reagent were added, and the tubes were closed and subjected to heating (100 °C) in a metabolic bath for 1 h. Once cooled to room temperature, the absorbance was read at 460 nm. The values were determined using a standard curve of catechin that was constructed (0–800 ppm). The results were expressed as catechin equivalents (CE)/g of the sample.

### 2.6. Phenolic Profile by High-Performance Liquid Chromatography Coupled to Mass Spectrometry (HPLC-MS)

The fermentative extracts were filtered through a 0.45 µm membrane and were collected in 1.5 mL vials to be placed in the HPLC equipment. The identification and characterization of the metabolites were determined in Varian HPLC equipment with an autosampler (Varian ProStar 410, Palo Alto, CA, USA), pump (Varian ProStar 2301, USA), and PDA detector (Varian ProStar 330, USA). A Denali C18 column (150 mm × 2.1 mm, 3 µm, Greace, USA) was used. The temperature was kept at 30 °C. Formic acid (0.2%) (Phase A) and acetonitrile (Phase B) were used as the mobile phase. The following gradient was used: initial, 3% B; 0–5 min, 9% B, linear; 5–15 min, 16% B, linear; 15–45 min, 50% B, linear. A flow of 0.2 mL/min was maintained. This equipment was coupled to an ion trap mass spectrometer (Varian 500 MS IT, USA) equipped with an electrospray ionization source. The ionization was carried out in negative mode. Nitrogen was used as the nebulizer gas, and helium was used as the buffer gas. The results of the polyphenolic compounds found in HPLC-MS were analyzed using Varian MS Workstation, Agilent Technologies, version 6.9 (Walnut Creek, CA, USA) [18].

### 2.7. Antioxidant Activity

The antioxidant activities of the extracts with higher HT and CT contents were determined using ABTS [19], DPPH [16], and FRAP [20], with some modifications.

#### 2.7.1. ABTS

A 7 mM ABTS (2, 2’-azino-bis (3-ethylbenzothiazoline-6-sulfonic acid)) solution was prepared, mixed with 2.45 mM K_2_S_2_O_8_, and left to incubate at room temperature for 12–16 h in the dark. The solution was diluted with ethanol until obtaining an absorbance of 0.7 ± 0.02 at 734 nm. Then, 190 μL of ABTS ethanolic solution and 10 µL of the extracts were mixed, allowed to react for 1 min, and read at 734 nm in a microplate reader (Epoch, Biotek, Instruments, Inc.). The ABTS solution with the solvent of the samples was taken as a control. Trolox was used as a standard, and the results were expressed as mg Trolox equivalents (TE)/100 g.

#### 2.7.2. DPPH

For DPPH, the radical was prepared at 60 µM in a methanolic solution. Then, 193 µL of this solution was mixed with 7 µL of the extracts in each well of a microplate. The mixture was allowed to react for 30 min at room temperature. The absorbance was read at 517 nm in a microplate reader (Epoch, Biotek, Instruments, Inc.). The DPPH solution with the solvent of the samples was taken as a control. Trolox was used as a standard, and the results were expressed as mg TE/100 g.

#### 2.7.3. FRAP

For the ferric reducing antioxidant power (FRAP) assay, the stock solutions were first prepared: TPTZ (10 mM) in HCl (40 mM), FeCl_3_6H_2_O (20 mM), and sodium acetate buffer (0.3 M, pH 3.6). The FRAP reagent was prepared at a ratio of 1:1:10 (TPTZ/FeCl_3_6H_2_O/acetate buffer). Subsequently, 10 μL of the sample was combined with 290 μL of FRAP solution in each well of a 96-well microplate. After 30 min, the absorbance was read at 593 nm in a microplate reader (Epoch, Biotek, Instruments, Inc.). The FRAP reagent with the solvent of the samples was taken as a control. Trolox was used as a standard, and the results were expressed as mg TE/100 g.

### 2.8. Statistical Analysis

All tests were performed by triplicate, except for the minerals, which were in duplicate. A one-way analysis of variance (ANOVA) followed by Tukey’s test (*p* ≤ 0.05) was performed to estimate the significant differences between the mean values in the proximal compositions and the mineral compositions of the two sorghum genotypes. Duncan’s test was used to evaluate the effect of the antioxidant activity of the fermentative extracts. Likewise, the Dunnett test was applied at a significance level of *p* ≤ 0.05 to compare the mean values of the tannin content of the fermented extracts at different fermentation times with respect to the control group of nonfermented extracts. The statistical analysis was performed using the SAS institute Inc., version 9.0 (CA, USA).

## 3. Results

### 3.1. Proximal Physicochemical Characterization

During the drying process, the moisture content was reduced in LES 5 to 12.43%, while in GB it was 12.78%. The WAI values found in LES 5 were 2.64, and in GB they were 2.65 g gel/g dry weight. In values obtained from the CHP tests, no significant differences were found (*p* ≤ 0.05) between the genotypes. The CHP values in both genotypes were below 40%; in LES 5 they were 36.47%, and in GB they were 36.72%.

The results of the proximal composition of two sorghum genotype grains are presented in Table 1. Carbohydrates were the macronutrients that represented the highest content, between 71.35% for LES 5 and 71.49% for GB; the carbohydrates in this cereal are composed mainly of starch. The protein contents were found to be between 10.50 and 10.89%. In lipids, the two genotypes contained 4.0–4.26%. The ash values were between 1.85 and 2.01%. Of the total carbohydrates, crude fiber was between 1.90 and 2.01%. In addition, Table 1 shows that sorghum is a rich source of minerals. In the two analyzed genotypes, significant differences were found (*p* ≤ 0.05). The highest potassium content value was obtained in GB (12.28 mg/g), followed by phosphorus in GB, with 2.29 mg/g. Calcium was another of the outstanding minerals; LES 5 had the highest content at 2.19 mg/g. GB had the highest chlorine content of 1.46 mg/g, and sulfur was also found in a significant amount in GB, with 1.19 mg/g. Other minerals found in smaller quantities were iron, manganese, zinc, and aluminum.

### 3.2. Solid-State Fermentation (SSF)—Assisted Extraction

For the radial growth assay and SSF, the initial pH values were 6.71 for LES 5 and 6.72 for GB. In Figure 1, the development of *A. niger* Aa210 on the substrate of the two sorghum genotypes was slightly lower than that of *A. oryzae*. Among the two analyzed strains, *A. oryzae* was the one that showed the highest growth speed, especially after 36 h in GB, so that between 60 and 72 h they managed to completely colonize the Petri dish. However, the growth analysis reflected similar behaviors between the two fungi since both allowed their development on the substrates at 72 h, as can be seen in Figure 1.

#### 3.2.1. Phenolic Content

In the comparison of HT of the fermentation times with the control, in the fermentative extracts of sorghum LES 5 and GB by *A. oryzae*, significant differences (*p* ≤ 0.05) were only found at 24 h of fermentation, where there were increases of 3.77 and 3.90 mg GAE/100 g, respectively. At the other times, the HT content was maintained during the fermentation process compared to the unfermented extracts, as shown in Table 2. In the fermentative extracts of LES 5 and GB by *A. niger* Aa210, significant increases (*p* ≤ 0.05) in the HT content were observed at 12, 24, 36, 48, 60, 72 and 96 h of fermentation compared to the unfermented extracts. Values were found from 3.90 to 5.90 GAE/100 mg (Table 2). Among the fermentation times of the HT variable, the highest values were observed at 72 h of fermentation by *A. niger* Aa210, which were 5.67 mg GAE/100 g for LES 5 and 5.90 mg GAE/100 g for GB (Table 2), yields corresponding to 70 and 84%, respectively, higher than the unfermented extract. Regarding the extracts of SSF by *A. oryzae*, significant differences were observed, including a greater decrease of 2% in HT in the extracts of LES 5 at 36 h (3.27 mg GAE/100 mg) of fermentation and a decrease in GB of up to 3% at 60 h (3.10 mg GAE/100 g) of fermentation (Table 2).

Regarding the content of CT, in the SSF extracts of LES 5 by *A. oryzae,* no significant differences were found. The results at the different fermentation times were lower compared to the unfermented extract, except at 12 h of fermentation (Table 3). However, significant decreases of 6 to 12% were observed at 36, 60, 72, 84 and 96 h of fermentation, with values between 45.20 and 48.03 mg CE/100 mg (Table 3). In the fermentative extracts of LES 5 by *A. niger* Aa210, significant increases (*p* ≤ 0.05) in CT were observed at 12, 48, 60, 72, 84 and 96 h of fermentation, and the highest content of CT was obtained at 72 h, which was 76.07 mg CE/100 g (Table 3), corresponding to 49% more than the unfermented extract. In the SSF extracts with GB and *A. oryzae*, significant decreases of 17 to 23% in CT were obtained at the fermentation times of 24, 36, 84 and 96 h (44.56–46.86 mg CE/100 g) compared to the unfermented extracts (Table 3). In the SSF extracts with GB by *A. niger* Aa210, compared with the unfermented extracts the contents of CT increased significantly at 36, 48, 72 and 84 h of fermentation (Table 3). In addition, a greater accumulation of CT was observed at 84 h of fermentation, which was 73.20 mg CE/100 g, whose yield was 33% more than in the unfermented extract.

#### 3.2.2. Phenolic Profile

The phenolic profiles of the SSF extracts of the LES 5 and GB genotypes with A. *oryzae* and *A. niger* Aa210 were determined by HPLC-MS at 0, 24, 48, 72, and 84 h of fermentation. These times were established based on the results obtained for the HT and CT contents, and a total of 21 compounds were identified (Table 4). Table 4 shows that there is a relationship between the tannin contents and the phenolic profiles studied in the fermentative extracts of SSF with *A. oryzae* and *A. niger* Aa210 in both genotypes of sorghum since a greater number of phenolic compounds were released in SSF by *A. niger* Aa210, while in SSF by *A. oryzae* the number of compounds released was lower. The compound (-)-epicatechin-(2a-7) (4a-8)-epicatechin 3-*O*-galactoside from the group of proanthocyanidin dimers, theaflavin 3,3’-*O*-digallate from the group of flavonoids, and other polyphenols such as arbutin remained unchanged in shape at certain times of fermentation and were found in most of the analyzed extracts. Of all the studied extracts, in those of SSF with GB and *A. niger* Aa210, a greater number of compounds were located at the different fermentation times: 10 at 24 h, 9 at 48 h, 9 at 72 h, and 6 at 84 h. Among the identified compounds were the phenols (+)-catechin, prodelphinidin trimer C-GC-C from the group of proanthocyanidin trimers, flavanones such as eriodictyol, (-)-epicatechin, and anthocyanins such as delphinidin 3-*O*-arabinoside, among others, which are shown in Table 4. Other condensed tannins were also identified as procyanidin trimer C1 (proanthocyanidin trimers), mainly in fermentative extracts of LES 5 with *A. niger* Aa210 and *A. oryzae*.

### 3.3. Antioxidant Activity

For this test, the extracts that exhibited the highest tannin content were selected, which were from SSF by *A. niger* Aa210 at 72 and 84 h of fermentation. All the extracts exhibited important antioxidant activity, as shown in Table 5. Significant differences were found, with the LES 5 extract fermented for 72 h showing the greatest inhibition of the ABTS radical, obtaining a value of 64.33 mg TE/100g. The highest DPPH radical scavenging activity was found with GB fermented for 84 h, which was 133.67 mg TE/100 g, while the biggest quantification of ferric reducing antioxidant power was registered with GB fermented for 84 h (59.23 mg TE/100 g) (Table 5). The CT content was related to the antioxidant activity by Pearson’s correlation analysis (*p* < 0.05); the CT values of the evaluated extracts correlated strongly and positively with the inhibitory activity of the ABTS radical (r = 0.962), while a positive media correlation was found with FRAP (r = 0.678). On the contrary, a weak positive correlation (r = 0.252) was demonstrated in the scavenging activity against the DPPH radical.

## 4. Discussion

### 4.1. Proximal Physicochemical Characterization

The purpose of drying is to reduce the moisture contents of foods by reducing water activity, thus obtaining a stable product with a longer shelf life. Drying involves a thermal treatment, and the variation in temperature can affect the degree of degradation of polyphenols. However, this effect also depends on the varieties and classes of polyphenols [21]. Lyophilization is considered the best drying technique for obtaining high-quality dry products for heat-sensitive compounds, although it is one of the most expensive methods. Convection drying is the most economical and widely used method in the food industry [22]. In this research, a drying temperature of 55 °C was selected since it has been shown in previous studies that polyphenolic compounds do not undergo degradation and may even increase their contents. Madrau et al. [23] evaluated the effects of two drying temperatures of 55 and 75 °C on the phenolic contents of two apricot cultivars. In one cultivar, the contents of two phenolic acids increased at the highest temperature, and in both cultivars, the catechin levels showed the same trend. In a study of cocoa beans, the drying effect of three temperature values, 60, 70 and 80 °C, on the contents of total phenols was analyzed, obtaining the maximum concentration at 70 °C, while the levels decreased as the temperature increased [24]. On the other hand, air-dried pomegranate peels dried at 40 and 60 °C contained higher phenolic contents than peels dried at 90 °C [25].

The humidity and the nature of the solid substrate are among the most important critical factors involved in SSF processes. Fungi generally need less humidity; for fungi growth, humidity levels of 40 to 60% are sufficient [9]. The WAI parameter indicates the ability of a sample to absorb water, which depends on the availability of hydrophilic groups that bind to water molecules and the gelling capacity of the macromolecules. Consequently, it is preferable to use materials with high WAI values because they facilitate the growth and development of microorganisms [26]. Therefore, the results showed sorghum to have good moisture absorption capacity. CHP corresponds to water bound to the support, which cannot be used by microorganisms for their metabolic activities. High CHP values constitute a smaller amount of water bound to the material, which can affect the development of microorganisms [27]. Substrates regularly have water contents that fluctuate between 30 and 85%, and lower levels can cause the sporulation of microorganisms, while higher values modify the porosity of the system [28]. A maximum limit of 40% has been recommended for the growth of *A. niger* in SSF [12]. Our results were below 40%. According to the results obtained in these physical tests, sorghum grain could be considered a suitable substrate to support fermentation in SSF processes.

The results obtained for the proximal composition of the sorghum grains are within those reported in the literature, with protein values from 5.4 to 12.9% [29], lipid values of 2.1–7.6%, crude fiber values of 1.03–3.4%, total carbohydrate values of 57–80.6%, and ash values of 1.3–3.5% [30]. It is important to mention that the nutritional composition of sorghum can be influenced, mainly by factors such as genotype and environment [31]. The kafirins are the main storage proteins in sorghum grain and account for 60% or more of the total protein [32]. Sorghum lipids are mainly made up of unsaturated fatty acids (83–88%). The main ones are linoleic 45.6–51.1%, oleic 32.2–42.0%, palmitic acid 12.4–16.0%, and linolenic 1.4–2.8% [33]. Sorghum grain is a rich source of minerals such as Ca, Fe, K, Mg, P, and Zn [34]. The substrate is the source of carbon and nutrients, which requires supplements for the optimal growth of microorganisms. Minerals that are generally added to the medium include phosphorus, sulfur, potassium, magnesium, calcium, zinc, manganese, copper, iron, cobalt, and iodine [35]. The selection of a suitable substrate is essential in SSF processes since the substrate will act as a physical support and a source of nutrients for the development of the microorganisms [36].

### 4.2. Solid-State Fermentation (SSF)—Assisted Extraction

The pH is one of the critical factors in a fermentation process. However, its control during fermentation is difficult. Although SSF has relative pH stability, an initial adjustment in the substrate pH is sufficient to eliminate the need for its control [37]. Therefore, the pH was measured at the beginning of the fermentation. The results obtained in the radial growth indicated that both strains of fungi could be capable of synthesizing metabolites of interest required for the adequate use of the nutrients of the substrates of the two sorghum genotypes.

#### 4.2.1. Phenolic Content

According to the results obtained for the phenolic content, in the decrease in HT in the extracts of the SSF of both genotypes of sorghum with the fungus *A. oryzae*, it has been reported that the loss of tannins during fermentation could be the result of tannase activity [38]. SSF represents an alternative culture system for the production of value-added products from microorganisms, especially enzymes or secondary metabolites. The *Aspergillus* fungus is one of the microorganisms considered a potential source of tannin acyl hydrolase, commonly called tannase, which is a hydrolytic enzyme [39] that catalyzes the hydrolysis of gallotannin and ellagitannin, producing gallic acid and ellagic acid, respectively [18]. Considering that hydrolyzable tannins are complex polymeric compounds and are classified into two subclasses—gallotannins when they come from gallic acid and ellagitannins when they are derived from ellagic acid, a dimer of gallic acid [40]—the decrease in HT could be attributed to the tannase activity produced by *A. oryzae*.

In accordance with Ritthibut et al. [41], in the fungal fermentation bioprocess, various enzymes with important lytic activities participate, targeting polymeric matrices are activated and induce the release of free phenolic compounds. In this regard, the increase in the content of hydrolyzable tannins in SSF by *A. niger* was due to the release of free phenolic compounds present in sorghum grains. Tanasković et al. [42] demonstrated that after SSF, the soluble phenolic content of wheat bran tripled compared to crude bran. Sadh et al. [43] found that the seeds and rice flour in SSF with *Aspergillus* strains whose samples were extracted with ethanol exhibited a higher total phenolic content, which was attributed to the high enzymatic activity of amylase, considering that the species *A. niger* and *A. oryzae* are important sources of hydrolytic enzymes such as α-amylases [44].

The contents of total polyphenols were different depending on the genotype. Previously reported studies found values from 0.046 mg GAE/g [45] to 3.38 mg GAE/g [46].

*A. niger,* used in this study, has been recognized for its high capacity to produce extracellular lipases, proteases, and cellulases, in addition to other hydrolytic enzymes such as amylases [47]. In this context, the hydrolysis exerted by the enzymes may have enhanced the release of condensed tannins, which have been reported to be cell-wall-bound [48].

The tannins have strong interactions with amylose, which contributes to the formation of resistant starch [49], and in addition, tannins reduce the digestibility of proteins, a fact that could be related to their ability to inhibit enzymes [50]. In this regard, this action may influence its greater resistance to the effect of the enzymes produced mainly by the fungus *A. oryzae* and thus complicates the hydrolytic capacity for the release of tannins. Therefore, the decrease in condensed tannins observed in SSF by *A. oryzae* is positive since they are considered antinutritional factors, so it is possible that during SSF the tannins have been hydrolyzed into low-molecular-weight compounds and may be more bioaccessible [51].

Sorghum is a rich source of procyanidins, representative compounds of the simplest condensed tannins in sorghum, the most common being the catechin and epicatechin units [52]. The main polyphenolic substances of sorghum are condensed tannins [53,54]. In this context, it could be considered that, for this reason, in this analysis they were found in higher concentrations in the presence of hydrolyzable polyphenols.

Ritthibut et al. [41] obtained an increase in bioactive compounds such as total phenolics, total flavonoids, and phenolic acids in SSF of rice bran with *Aspergillus* strains. Wen et al. [55] reported a decrease in tannin content after the SSF of four common Chinese herbal medicinal residues with *A. oryzae*. This reduction could be due to enzymatic systems other than tannase. Likewise, it has been reported that condensed tannins can degrade more slowly compared to hydrolyzable tannins [55]. In previously reported studies with extracts of pigmented sorghum grains, various values of condensed tannins have been found, such as 0.0144 mg CE/g [56], 0.41 mg CE/g [51,57], 6.5 mg CE/g [58], and 6.7 mg CE/g [59]. It is also important to mention that the sorghum genotype and the environment of the crop affect its color, appearance, and nutritional quality, which contributes to the differences found in the tannin contents [31].

#### 4.2.2. Phenolic Profile

Sorghum grain contains phenolic acids, which are mainly found in the pericarp, testa, aleurone layer, and endosperm [31]. In the present study, two hydroxycinnamic acids were found, caffeic acid 4-*O*-glucoside and p-coumaroyl tyrosine, as well as a hydroxybenzoic acid, galloyl glucose. Wu et al. [4] identified a total of 25 individual polyphenols in methanolic extracts of sorghum grains by HPLC coupled with diode array detection and electrospray ionization mass spectrometry, including the compound eriodictyol. Eleven phenolic compounds in aqueous extracts from condensed tannin sorghum bran were reported, including caffeic acid and eriodictyol, which were identified by HPLC/MS [59].

Dykes et al. [60] studied the flavonoid compositions of 13 varieties of sorghum with red pericarps using HPLC-DAD. The compound eriodictyol was one of the most predominant flavanones in two of the analyzed varieties. Most of the monomeric flavonoids present in sorghum have a phenol or catechol group in the B-ring, so they are derived from naringenin (apigenin, apigeninidin, and derivatives of proapigeninidin) or eriodictyol (luteolin, luteolinidin, proluteolinidin, and procyanidin derivatives) in the phenylpropanoid pathway [61].

The main monomer of tannins in sorghum grain is (+)-catechin, which serves as a terminator of the tannin chain, while the monomer (-)-epicatechin can act as a termination and extension unit [62]. In this study, we report the presence of polymeric proanthocyanidins such as prodelphinidin trimer C-GC-C, procyanidin trimer C1, procyanidin trimer C2, and (-)-epicatechin-(2a-7) (4a-8)-epicatechin 3-*O*-galactoside. Lignans possess significant antioxidant capacity and are considered sources of phytoestrogens in the diet [63]. In this analysis, the phenol medioresinol of the lignan family was identified.

The contents and phenolic profiles in sorghum differ since it is influenced by the variety, genotype, and pericarp color as well as the presence of a testa and whether it is pigmented or unpigmented [64].

### 4.3. Antioxidant Activity

ABTS and DPPH are two free radicals that are commonly used to evaluate antioxidant activity in vitro. The ABTS method is fast and can be used over a wide range of pH values in both aqueous and organic solvent systems [65]. The DPPH method is widely used to determine the antiradical/antioxidant activity of both purified phenolic compounds and natural plant extracts [1]. While the ABTS and DPPH methods are discoloration-based assays, the quantification of FRAP is based on the increase in absorbance at a specific wavelength that occurs when antioxidant compounds react with a chromogenic reagent [66].

Sorghum has a high antioxidant activity, provided mainly by phenolic acids, condensed tannins, and anthocyanins [67]. Sorghum varieties that contain tannins have the strongest antioxidant properties in vitro, which is attributed to a greater free radical scavenging power of tannins relative to simple flavonoids. The hydroxyl groups of condensed tannins are close to each other and are therefore more effective in scavenging peroxyl radicals than simple phenolics [68]. Preliminary studies have reported that pigmented sorghum extracts have strong antioxidant activities. Punia et al. [56] evaluated the antioxidant activity in a variety of red sorghum and obtained results of 20.55, 45.66, and 15.34 mg TE/100 g by the DPPH, ABTS, and FRAP methods, respectively. Rao et al. [69], in pigmented sorghum varieties, reported high antioxidant activity, with values of 20.92 and 21.02 mg TE/g on the FRAP and ABTS assays, respectively. Sorghums containing condensed tannins have shown higher antioxidant activities in vitro and are comparable to the antioxidant levels of fruits and vegetables [70].

On the other hand, the differences found in the Pearson’s correlation analysis could be due to the structures of the different compounds found in the evaluated extracts [71]. Previous research has shown that the tannin content and total phenolic content of sorghum are strongly correlated with antioxidant activity [31,72,73,74].

## 5. Conclusions

The efficacy of SSF in obtaining bioactive compounds was demonstrated since, with the fungus *A. niger* A210, a greater release of hydrolyzable and condensed tannins was obtained. The SSF of sorghum substrates with *A. oryzae* favored the reduction of the phenolic content in such a way that sorghum could be used in human diets or the fortification of other foods since the condensed tannins in sorghum represent a limiting factor for its use. In the phenolic profile, 21 compounds were identified. A greater number of phenolic compounds were located in SSF by *A. niger* Aa210, while in SSF by *A. oryzae* the number of released compounds was lower. Likewise, the antioxidant capacity of the sorghum extracts fermented by *A. niger* after 72 and 84 h fermentation periods was demonstrated since they exhibited a significant inhibition capacity on ABTS and DPPH radicals and a greater quantification of FRAP, which is positive considering that in sorghum the compounds with higher levels of antioxidants are tannins. Therefore, these extracts could be used as antioxidant agents in different food products.

## Figures and Tables

**Figure 1 foods-11-03121-f001:**
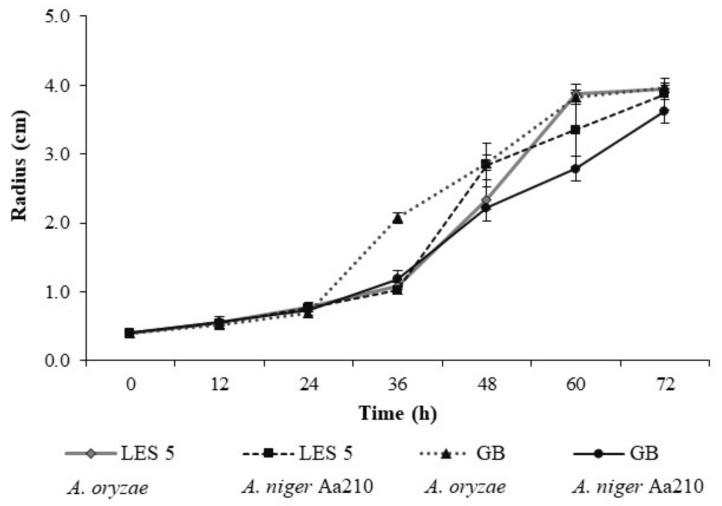
Radial growth (cm) of two fungi, *A. oryzae* and *A. niger* Aa210, on the substrates of two sorghum genotypes, LES 5 and GB, at 72 h.

**Table 1 foods-11-03121-t001:** Proximate composition (%) dry weight and mineral composition (mg/g) of sorghum grains of two genotypes.

Parameter	LES 5	GB	Mineral	LES 5	GB
Moisture	9.66 ± 0.34 ^a^	10.10 ± 0.48 ^a^	K	11.17 ± 0.04 ^b^	12.28 ± 0.01 ^a^
Total carbohydrates	71.35 ± 0.15 ^a^	71.49 ± 0.02 ^a^	P	1.99 ± 0.01 ^b^	2.29 ± 0.01 ^a^
Protein	10.89 ± 0.34 ^a^	10.50 ± 0.00 ^a^	Ca	2.19 ± 0.11 ^a^	1.95 ± 0.01 ^a^
Lipids	4.26 ±0.22 ^a^	4.00 ± 0.40 ^a^	Cl	1.27 ± 0.04 ^b^	1.46 ± 0.00 ^a^
Ash	1.85 ± 0.02 ^b^	2.01 ± 0.02 ^a^	S	1.00 ± 0.07 ^a^	1.19 ± 0.00 ^a^
Crude fiber	2.01 ± 0.13 ^a^	1.90 ± 0.07 ^a^	Fe	0.23 ± 0.01 ^b^	0.28 ± 0.00 ^a^
			Mn	0.11 ± 0.00 ^a^	0.12 ± 0.01 ^a^
			Zn	0.10 ± 0.01 ^b^	0.15 ± 0.00 ^a^
			Al	0.06 ± 0.00 ^b^	0.13 ± 0.02 ^a^

The data represent the means ± standard deviations of the corresponding experiments. Different letters indicate significant differences according to Tukey’s multiple comparison test (*p* ≤ 0.05).

**Table 2 foods-11-03121-t002:** Average hydrolyzable tannins (HT) in extracts of LES 5 and GB sorghum during solid-state fermentation (SSF) by *A. oryzae* and *A. niger* Aa210. Time 0 = unfermented extract.

	HT (mg GAE/100 g)			
Fermentation Time (h)	LES 5 SSF by *A. oryzae*	LES 5 SSF by *A. niger* Aa210	GB SSF by *A. oryzae*	GB SSF by *A. niger* Aa210
0	3.33 ± 0.12	3.33 ± 0.12	3.20 ± 0.12	3.20 ± 0.12
12	3.73 ± 0.06	4.23 ± 0.06 *	3.30 ± 0.06	4.27 ± 0.15 *
24	3.77 ± 0.12 *	3.97 ± 0.06 *	3.90 ± 0.58 *	3.90 ± 0.10 *
36	3.27 ± 0.12	4.27 ± 0.06 *	3.40 ± 0.23	4.50 ± 0.00 *
48	3.53 ± 0.15	4.90 ± 0.27 *	3.20 ± 0.06	4.80 ± 0.12 *
60	3.40 ± 0.20	5.33 ± 0.06 *	3.10 ± 0.10	4.90 ± 0.06 *
72	3.57 ± 0.12	5.67 ± 0.12 *	3.30 ± 0.00	5.90 ± 0.10 *
84	3.70 ± 0.46	3.47 ± 0.12	3.40 ± 0.06	3.40 ± 0.25
96	3.53 ± 0.23	4.30 ± 0.00 *	3.20 ± 0.44	4.70 ± 0.00 *

* Represents *p* ≤ 0.05 compared with the unfermented extract according to Dunnett’s multiple comparison test.

**Table 3 foods-11-03121-t003:** Average condensed tannins (CT) in extracts of LES 5 and GB sorghum during solid-state fermentation (SSF) by *A. oryzae* and *A. niger* Aa210. Time 0 = unfermented extract.

	CT (mg CE/100 g)			
Fermentation Time (h)	SSF LES 5 by *A. oryzae*	SSF LES 5 by *A. niger* Aa210	SSF GB by *A. oryzae*	SSF GB by *A. niger* Aa210
0	50.90 ± 5.15	50.90 ± 5.15	54.90 ± 3.12	54.90 ± 3.12
12	55.30 ± 3.05	61.63 ± 2.47 *	50.36 ± 1.33	59.96 ± 3.43
24	49.50 ± 2.75	59.97 ± 6.24	46.86 ± 1.33 *	59.76 ± 3.72
36	47.13 ± 1.96	55.30 ± 3.72	45.20 ± 2.95 *	67.50 ± 5.38 *
48	49.73 ± 4.57	72.27 ± 4.47 *	50.50 ± 1.45	68.30 ± 2.18 *
60	48.03 ± 0.40	70.70 ± 5.15 *	55.30 ± 3.32	63.40 ± 3.37
72	48.00 ± 3.29	76.07 ± 3.50 *	51.90 ± 3.27	67.70 ± 5.20 *
84	45.20 ± 1.14	64.30 ± 4.59 *	44.56 ± 2.65 *	73.20 ± 5.49 *
96	46.33 ± 3.16	69.03 ± 3.54 *	44.90 ± 2.40 *	62.50 ± 5.26

* Represents *p* ≤ 0.05 compared with the unfermented extract according to Dunnett’s multiple comparison test (*p* ≤ 0.05).

**Table 4 foods-11-03121-t004:** Polyphenolic compounds identified by HPLC-MS in grain extracts of two sorghum genotypes in SSF by *A. oryzae* and *A. niger* Aa210 after 0, 24, 48, 72, and 84 h of fermentation. Time 0 Ue = unfermented extract.

No.	Dough	Compound	Family	LES 5 SSF			GB SSF		
				Ue	*A. oryzae*	*A. niger* Aa210	Ue	*A. oryzae*	*A. niger* Aa210
1	357.1	Gardenin B	Methoxyflavones		24, 48				
2	865.1	Procyanidin trimer C1	Proanthocyanidin trimers	0	24, 48, 84	24, 48, 72	0		24
3	864.1	Procyanidin trimer C2	Proanthocyanidin trimers	0	24				
4	867.1	Theaflavin 3,3’-*O*-digallate	Flavonoids	0	24, 48, 72,84	24, 72	0	24, 48, 72, 84	24, 48, 72
5	705.2	(-)-Epicatechin-(2a-7)(4a-8)-epicatechin 3-*O*-galactoside	Proanthocyanidin dimers	0	24, 48, 72, 84	24, 48, 72, 84		48, 72, 84	24, 48, 72, 84
6	285.0	Scutellarein	Flavones	0	24, 48, 72, 84	24, 48, 72			24
7	271.0	Arbutin	Other polyphenols		24, 84	24, 84	0	24, 48, 72	24, 48, 72
8	329.1	3,7-Dimethylquercetin	Methoxyflavonols		24, 48				
9	289.0	(+)-Catechin	Catechins	0	48	24	0	48	24, 48, 72
10	330.8	Galloyl glucose	Hydroxybenzoic acids		72, 84			24, 48, 72	24
11	883.1	Prodelphinidin trimer C-GC-C	Proanthocyanidin trimers	0			0	24	24, 48, 72
12	287.0	Eriodictyol	Flavanones			72, 84		24	24, 72, 84
13	716.1	Theaflavin 3’-*O*-gallate	Theaflavins					72	
14	341.0	Caffeic acid 4-*O*-glucoside	Hydroxycinnamic acids			48, 72, 84	0	84	48, 72, 84
15	377.0	3,4-DHPEA-EA	Tyrosols	0		72, 84	0	84	48, 84
16	272.9	Phloretin	Dihydrochalcones			24			24
17	415.1	Daidzin	Isoflavones	0					
18	327.2	p-Coumaroyl tyrosine	Hydroxycinnamic acids				0		
19	289.0	(-)-Epicatechin	Catechins			72, 84			48, 72, 84
20	387.1	Medioresinol	Lignans			84			
21	434.1	Delphinidin 3-*O*-arabinoside	Anthocyanins						48, 72, 84

**Table 5 foods-11-03121-t005:** Antioxidant activities of fermentative extracts of two sorghum genotypes, LES 5 and GB, determined by the ABTS, DPPH, and FRAP assays.

Extract	Fermentation Time (h)	ABTS (mg TE/100 g)	DPPH (mg TE/100 g)	FRAP (mg TE/100 g)
LES 5	72	64.33 ± 1.15 ^a^	126.67 ± 1.15 ^b^	54.00 ± 3.17 ^ab^
	84	61.67 ± 0.58 ^b^	127.67 ± 0.58 ^b^	50.20 ± 2.25 ^bc^
GB	72	62.33 ± 0.58 ^b^	127.00 ± 1.00 ^b^	47.47 ± 1.75 ^c^
	84	63.00 ± 1.00 ^ab^	133.67 ± 1.15 ^a^	59.23 ± 3.91 ^a^

Different letters indicate significant differences according to Duncan’s multiple comparison test (*p* ≤ 0.05).

## Data Availability

No data reported.

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
