# Peer review of "Solid-State Fermentation of Sorghum by Aspergillus oryzae and Aspergillus niger: Effects on Tannin Content, Phenolic Profile, and Antioxidant Activity"

_foods, 2022, doi:10.3390/foods11193121_

Round 1

Reviewer 1 Report

I have reviewed the manuscript titled: Solid-state fermentation of sorghum by Aspergillus oryzae and Aspergillus niger: Effects on tannin content, phenolic profile, and antioxidant activity.

This article aims to evaluate the effect of solid-state fermentation from Aspergillus niger Aa210 and Aspergillus oryzae on phenolic profile, tannin, and antioxidant activity from two pigmented sorghum. The effects of Aspergillus niger in the yield of hydrolyzable tannins and condensed tannins was 70-84% and 33-49%, respectively, which was higher than those of Aspergillus oryzae. The extracts fermented by Aspergillus niger exhibited a higher antioxidant activity than that by Aspergillus orzyae. Therefore, Aspergillus niger was considered as an efficient fungus for the release of phenolic ompounds that can be used to improve the bioavailablity and increase antioxidant activity of food products through sorghum solid state fermentation. It also could be used for the by product of distilled sorghum liquor for further usage after sorghum solid-state fermentation and handle disposal problems. And the antioxidant capacity of the sorghum extracts fermented by Aspergillus niger at 72-84 hour fermentation periods was shown had a significant inhibition capacity of DPPH radicals, ABTS, and a greater quantification of FRAP for different food product addition. The article is innovative and it contains original and interesting information for sorghum solid-state fermentation data for cereal processor and solid-state fermentation liquor manufacture. Abstract is well written upon and 21 polyphenols were mentioned in the extracts. Introduction is well addressed including sorghum is the fifth most important cereal in the world and sorghum is a gluten-free cereal for celiac disease patient. The bioactive polyphenolic compounds in sorghum such as phenolic acids, tannins, and flavonoids, which could contribute to the antioxidant capacity of food components but also an antinutritional factors, which interfere with the digestibility of carbohydrates and proteins even mitigating the absorption of nutrients. The importance of cereal processing chooses enhancing or decreasing bioactive compounds concentrations and modifiy the process could be decided. The mitigating release of phenolic compounds from sorghum grains by solid-state fermention and decrease the content of tannins by Aspergillus oryzae could be adopted.

Materials and methods were well described. However, the AOAC methods (1980) for protein, ash and fiber content measurement were not cited in reference section. Two abbreviated TH instead of hydrolyzable tannins (HT) were missed spelled in section 3.2.1 Phenolic content of Results section, which needed for revising.

This article would be accepted if the authors revise the Journal name abbreviation of papers in reference section. It will be helpful for other researchers to follow this study in the future.

I am not a native English speaker. The manuscript seems have no major mistakes are detected and the manuscript can be easily understood as attached file. The results are well discussed.

I enjoyed reading this manuscript; the needs of special groups of sorghum grain processing and sorghum solid-state fermentation for liquor manufacture. This manuscript presents some interesting data.

Date of this review

11 September 2022 10:19 pm

Author Response

  1. Materials and methods were well described. However, the AOAC methods (1980) for protein, ash and fiber content measurement were not cited in reference section.

Response: According to the observation, it is already cited in number 13 of the list of references.

  1. Two abbreviated TH instead of hydrolyzable tannins (HT) were missed spelled in section 3.2.1 Phenolic content of Results section, which needed for revising.

Response: The observation corrections are found on lines 299, 302, and 311.

  1. This article would be accepted if the authors revise the Journal name abbreviation of papers in reference section. It will be helpful for other researchers to follow this study in the future.

Response: According to the observation, the abbreviations of the journals in the reference list have already been corrected.

Reviewer 2 Report

Line 22- Please change the sentence to: In the extracts 21 polyphenols were identified....

Line 97- For the WAI 1.5 g of sorghum samples were.....

Line 152-. Briefly.....

Line 214- Why duplicates in mineral tests if the other test were all done in triplicates?

Line 235- Table 1

Line 272- Figure 1

Line 302- Table 2

Have a great day!

Author Response

  1. Line 22- Please change the sentence to: In the extracts 21 polyphenols were identified....

Response: The change in line 22 is done

  1. Line 97- For the WAI 1.5 g of sorghum samples were.....

Response: The change was made in lines 101 and 102 because the space was moved.

  1. Line 152-. Briefly.....

Response: The change was made on line 156.

  1. Line 214- Why duplicates in mineral tests if the other test were all done in triplicates?

Response: Because the epsilon 1 x-ray fluorescence analyzer is a fast, highly precise, and accurate instrument the results are practically the same from one replica to another, that is why a third replica was not necessary.

  1. Line 235- Table 1

Response: This misspelling has been corrected on line 241.

  1. Line 272- Figure 1

Response: This misspelling has been corrected on line 272.

  1. Line 302- Table 2

Response: This misspelling has been corrected on line 303.

Reviewer 3 Report

Authors present the assessment of sorghum solid-state fermentation effect on bioactive components. The manuscript is appropriately written, and it contains relevant contributions to food technologies. However, my recommendation is to accept publication after the following minor revisions:

Typo should be revised. (Line 63)

Concentrations of all reagents should be provided.

It would be interesting to explain why 55°C temperature and 48h were chosen to dry samples as they can affect bioactive compounds.

It would be interesting to explain if all tests were done with random sequence.

It would be interesting to add if ANOVA assumptions were guaranteed (homogeneity of variance, independent residuals and normal distribution of residuals).

Author Response

  1. Typo should be revised. (Line 63)

Response: The typo has been corrected on line 63.

  1. Concentrations of all reagents should be provided.

Response: We agree and are mentioned in section 2.1. Genetic material, microorganisms, and chemicals.

  1. It would be interesting to explain why 55°C temperature and 48h were chosen to dry samples as they can affect bioactive compounds.

Response: We agree with the suggestions, therefore the moisture content is mentioned in the results section at the beginning of section 3.1 on lines 231 and 232; while in the discussion a paragraph was added at the beginning of section 4.1 from lines 402 to 418, whose citations are found in the list of references.

  1. It would be interesting to explain if all tests were done with random sequence.

Response: We agree with your suggestion, however, in the statistical analysis section it is addressed that in a randomized trial there is a direct comparison between two or more treatment groups, one of which serves as a control, such is the case of the Dunnett's test when comparing the content of tannins in the fermentative extracts against the level of tannins in the unfermented extracts.

  1. It would be interesting to add if ANOVA assumptions were guaranteed (homogeneity of variance, independent residuals, and normal distribution of residuals).

Response: A one-way analysis of variance (ANOVA) was warranted which we added in the statistical analysis and the standard deviations of the mean values were calculated, however, no normal distribution test was performed.